# Does varying the ingestion period of sodium citrate influence blood alkalosis and gastrointestinal symptoms?

**Charles S. Urwin**[1]*, **Rodney J. Snow**[2], **Liliana Orellana**[3], **Dominique Condo**[1,2], **Glenn D. Wadley**[2], **Amelia J. Carr**[1]

**1** Centre for Sport Research, School of Exercise and Nutrition Sciences, Faculty of Health, Deakin University, Geelong, Victoria, Australia, **2** Institute for Physical Activity and Nutrition, School of Exercise and Nutrition Sciences, Faculty of Health, Deakin University, Geelong, Victoria, Australia, **3** Biostatistics Unit, Faculty of Health, Deakin University, Geelong, Victoria, Australia

* urwinc@deakin.edu.au

## Abstract

### Objectives

To compare blood alkalosis, gastrointestinal symptoms and indicators of strong ion difference after ingestion of 500 mg.kg$^{-1}$ BM sodium citrate over four different periods.

### Methods

Sixteen healthy and active participants ingested 500 mg.kg$^{-1}$ BM sodium citrate in gelatine capsules over a 15, 30, 45 or 60 min period using a randomized cross-over experimental design. Gastrointestinal symptoms questionnaires and venous blood samples were collected before ingestion, immediately post-ingestion, and every 30 min for 480 min post-ingestion. Blood samples were analysed for blood pH, [HCO$_3^-$], [Na$^+$], [Cl$^-$] and plasma [citrate]. Linear mixed models were used to estimate the effect of the ingestion protocols.

### Results

For all treatments, blood [HCO$_3^-$] was significantly elevated above baseline for the entire 480 min post-ingestion period, and peak occurred 180 min post-ingestion. Blood [HCO$_3^-$] and pH were significantly elevated above baseline and not significantly below the peak between 150–270 min post-ingestion. Furthermore, blood pH and [HCO$_3^-$] were significantly lower for the 60 min ingestion period when compared to the other treatments. Gastrointestinal symptoms were minor for all treatments; the mean total session symptoms ratings (all times summed together) were between 9.8 and 11.6 from a maximum possible rating of 720.

### Conclusion

Based on the findings of this investigation, sodium citrate should be ingested over a period of less than 60 min (15, 30 or 45 min), and completed 150–270 min before exercise.

**Data Availability Statement:** The dataset can be accessed via the following: DOI: 10.26187/tkah-h625 URL: http://hdl.handle.net/10536/DRO/DU:30149426.

**Funding:** CU received all funding for this investigation from the School of Exercise and Nutrition Sciences at Deakin University. The funders had no role in study design, data collection and analysis, decision to publish, or preparation of the manuscript.

**Competing interests:** The authors have declared that no competing interests exist.

## Introduction

Athletes competing in events of short-duration and high-intensity can limit the potentially deleterious effects of blood acidosis by ingesting buffering agents (such as sodium citrate and sodium bicarbonate) that induce blood alkalosis (a significant increase in blood pH and blood bicarbonate concentration ($[HCO_3^-]$)) prior to exercise [1–3]. These dietary supplements have been reported to induce some gastrointestinal (GI) disturbances [4–6], but it has been proposed that sodium citrate may induce fewer GI symptoms than sodium bicarbonate [7, 8]. A recent investigation identified that sodium citrate was indeed associated with reduced GI disturbances compared to sodium bicarbonate when the supplements were ingested at the same dose (300 mg.kg$^{-1}$ BM) [8]. These findings provide preliminary evidence that sodium citrate may be a preferred alkalising agent from a GI disturbance perspective.

While buffering agent ingestion is typically undertaken with the intent of improving subsequent exercise performance, equivocal effects on exercise performance have been reported after sodium citrate supplementation [9–11]. A recent review identified that an ergogenic benefit was more frequently reported when completing short-duration ($> 60$ s and $< 420$ s) and very high-intensity exercise ($> 100\%$ $VO_{2max}$) compared to longer-duration exercise of any intensity [12]. Performance of an all-out or high-intensity exhaustive effort at the end of endurance exercise has also been reportedly improved by supplementation with sodium citrate [13, 14]. Other equivocal performance outcomes may be partially attributed to sub-optimal ingestion protocols, where supplementation failed to maximise blood alkalosis or was associated with excessive GI symptoms prior to the commencement of exercise, relative to other ingestion protocols [12].

Based on the induced blood alkalosis and gastrointestinal symptoms reported in prior dose-response investigations, sodium citrate is recommended to be ingested at a dose of 500 mg.kg$^{-1}$ body mass (BM) [4, 15]. Ingestion in gelatine capsules rather than in solution is recommended to maximise blood alkalosis and palatability (i.e. participant preference) [16]. While sodium citrate dose and ingestion mode are somewhat established, no prior investigation has assessed the effect of the ingestion period (i.e. the time taken to complete ingestion of the entire dose of the supplement) on subsequent blood alkalosis, GI symptoms or palatability. The combined effect of a specific dose, mode and period of sodium citrate supplementation may also contribute to the suggested timing of ingestion (relative to the onset of exercise) required for performance benefit. Currently, sodium citrate supplementation is recommended to take place at least 180 min before the onset of exercise [4, 16], but changed physiological responses, palatability or GI symptoms according to the duration of the ingestion period have yet to be established.

A 30 min ingestion period has been most frequently implemented in sodium citrate investigations [4, 17–20], with blood alkalosis regularly induced. Shorter (5 to 10 min [13, 21–24]) and longer (90 to 150 min [25–27]) ingestion periods have also been trialled across investigations, although both extremes present challenges in practice. Short ingestion periods may not allow sufficient time to ingest the number of capsules required to meet the recommended dose (~ 36 capsules per session in a prior investigation [16]), while longer periods could incur greater disruption to athletes' event preparation. The extent to which a particular sodium citrate ingestion period is feasible for athlete implementation may also vary according to GI symptoms and palatability.

No prior sodium citrate investigation has monitored blood alkalosis for more than 240 min post-ingestion, with previously reported maximum alkalosis values occurring between 180 and 210 min after ingestion [4, 16]. Further examination of post-ingestion blood alkalosis for more than 240 minutes may allow more precise recommendations regarding the timing of

sodium citrate supplementation in relation to the commencement of exercise. It has been proposed that a 6 mmol.L$^{-1}$ increase in blood [HCO$_3^-$] following buffering agent ingestion may increase the likelihood of observing an ergogenic benefit [11, 28, 29]. However, it is unknown for how long elevated blood [HCO$_3^-$] remains above the 6 mmol.L$^{-1}$ threshold following sodium citrate ingestion, adding to the requirement for an extended post-ingestion observation period. Extending this post-ingestion observation period to 480 min would double the time explored by prior investigations, and may be sufficient to observe the full time interval where blood alkalosis remains elevated following sodium citrate supplementation.

Sodium citrate supplementation likely induces blood alkalosis via alterations to strong ion difference (SID) [30, 31], but this has yet to be directly established. Changes in blood sodium ([Na$^+$]), blood chloride ([Cl$^-$]) and plasma citrate concentrations ([citrate]) can represent the balance between the rate of entry into and removal from the circulation of these ions following sodium citrate supplementation, providing an indication as to changes in SID.

The primary aim of this investigation was therefore to examine the effect of four different ingestion periods (15, 30, 45 and 60 min) of 500 mg.kg$^{-1}$ BM sodium citrate on blood alkalosis (peak and time to peak for blood pH and [HCO$_3^-$]) over an extended (480 min) post-ingestion period. Secondary aims were to establish the effect of sodium citrate ingestion period on blood [Na$^+$], blood [Cl$^-$], plasma [citrate], GI symptoms, and palatability.

## Methods

### Participants

Healthy participants ($n$ = 16; 8 males and 8 females; mean ± standard deviation for age, 25.4 ± 4.7 years; body mass, 68.3 ± 12.7 kg; height, 1.73 m ± 0.11 m; VO$_{2peak}$, 46.0 ± 6.0 mL.kg.min$^{-1}$) were recruited. Eligibility was assessed via a health status questionnaire, with individuals excluded if they reported any history of kidney disease or the use of medication that can modify blood acidity regulation. Participants were informed verbally and in writing of the nature of the investigation, including potential risks, and signed a written informed consent statement prior to testing. The Deakin University Human Research Ethics Committee approved all protocols (2018–257).

### Study design overview

For each participant, assessment of maximal aerobic capacity (VO$_{2peak}$), height and body mass was completed a minimum of three days prior to the first experimental testing session. VO$_{2peak}$ was determined as previously described by Urwin et al. [16], using an incremental increase in cycling intensity to volitional fatigue. These respiratory data were used to categorise participants as "healthy" or "physically fit" according to previously used nomenclature [32]. Participants were randomly allocated to a sequence of four ingestion periods (15, 30, 45 and 60 min ingestion periods) using a counter-balanced, crossover design. On the two days immediately before the first session, 24-hour food and activity diaries were completed by participants, with details of all food and fluid ingested, as well as the type, intensity and duration of exercise completed. On the day prior to the first session, eight of sixteen participants collected all excreted urine over an eight hour period in a provided container (pre-testing urine), which was kept refrigerated (at approximately 4˚C), and delivered to the researcher upon arrival at the first session. Participants were required to commence the urine collection period immediately after voiding the bladder, starting at the same time of day as the subsequent sessions (e.g. 8:00 am). Participants avoided alcohol consumption and standardised their caffeine consumption for the 24 hours prior to each session, with adherence checked by a member of the research team. Participants arrived at the laboratory following an overnight fast that commenced at 10:00 pm

the night before each session. The mean (± SD) number of days between sessions (washout period) was 5.9 (± 2.8) days.

## Experimental testing sessions

Participants ingested 500 mg.kg$^{-1}$ BM sodium citrate (34.2 ± 6.3 capsules per participant) in size 0 gelatine capsules (Melbourne Food Ingredient Depot, Melbourne, Australia) with 750 mL of a chilled sports drink (Powerade, Coca Cola, USA), prepared according to manufacturer instructions. Ingestion periods between sessions were 15, 30, 45 or 60 min. During the ingestion period, participants also consumed a standardised meal, comprising 1.75 g.kg$^{-1}$ BM of carbohydrate [33–36], including 750 mL of sports drink, bread, jam, bananas and muesli bars. According to manufacturer packaging, the co-ingested meals (including Powerade) contained a total of approximately 2 g of sodium, representing 18% of the total ingested sodium (given that approximately 9 g sodium was ingested as sodium citrate). The amount of citrate in the co-ingested meal is unknown as it is not listed on the manufacturers packaging nor included in Australian food databases due to the small amounts in the food supply, therefore the amount is likely negligible.

To ensure standardised ingestion, the capsules, drinks and meals were provided to participants in three equal portions throughout the ingestion period; at commencement, halfway through, and at the end. Participants remained seated for the next 480 min, and a second meal was provided 240 min after completion of sodium citrate ingestion, comprising the same foods, quantities and carbohydrate content as the initial co-ingested meal.

## Tissue collection and analysis

Blood samples were collected via a cannula inserted in the antecubital vein. At baseline, immediately following the completion of ingestion, and then every 30 min for 480 min following the completion of ingestion, 1 mL of blood was drawn into a safePICO syringe (Radiometer, Copenhagen, Denmark), which was immediately analysed for blood pH, [$HCO_3^-$], [$Na^+$] and [$Cl^-$] using an ABL800Flex Blood-Gas Analyser (Radiometer, Copenhagen, Denmark). Another 4 mL of blood was drawn into a lithium-heparin coated vacutainer tube (McFarlane Medical, Melbourne, Australia) which was immediately centrifuged at 1,300 rpm for 10 min at 4˚C, after which the plasma was stored at -80 ˚C. Subsequently, the plasma (100 µL) was deproteinised in 50 µL of 1.5 M perchloric acid and neutralised by addition of 37 µL of 2.1 M $KHCO_3$ before being analysed for [citrate] using an enzymatic fluorometric assay [37, 38]. The intra-assay percentage coefficient of variation (%CV) for a mid-range citrate standard solution (i.e. 250µM) was 3.4%, and the inter-assay %CV was 10.3%.

Participants voided their bladder immediately before beginning sodium citrate ingestion. For the duration of each session (from the commencement of sodium citrate ingestion to 480 min post-ingestion), all excreted urine was collected and the volume (mL) assessed using a measuring cylinder. After mixing, a 1 mL aliquot of urine was obtained and stored at -80 ˚C until analysed for urinary [citrate] using the same method as for plasma. From urine volume and urinary [citrate] (µmol.L$^{-1}$), total excreted citrate (mg) was calculated.

## Validated questionnaires/scales

Participants completed GI symptoms questionnaires [39] at the same time-points as blood sampling. The severity of 10 GI symptoms (nausea, vomiting, bloating, abdominal cramps, early satiety, heartburn, sickness, loss of appetite, retrosternal discomfort, and upper abdominal pain) were rated on a 5-point Likert-type scale from 0 = no problem, to 4 = very severe problem. Palatability was quantified using a scale of participant preference immediately after

completion of ingestion [40], with participants rating the extent to which they liked ingesting sodium citrate on a 9-point Hedonic scale from 1 = dislike extremely to 9 = like extremely.

## Statistical analyses

Longitudinal measurements of blood variables were assessed using linear mixed models (LMM) including participant as a random effect, and fixed effects: treatment (ingestion period, four levels), time as a categorical variable (18 levels; baseline and 0 to 480 min post-ingestion at 30 min intervals), the interaction treatment by time and the order in which each treatment was administered. These LMM estimates were used to obtain the peak (gPeak; the maximum mean value as ascertained from cumulative group data, rather than for each participant in each individual session); time to gPeak (min; the time interval from the completion of ingestion to gPeak); and the time interval where mean values were significantly different from baseline but not significantly different from gPeak. This interval was estimated simultaneously for all treatments when the interaction was non-significant but also for the four treatments individually as an exploratory analysis. The same approach was used for minimum blood [Cl⁻] (gMin, time to gMin). A blood [HCO₃⁻] increase of 6 mmol.L⁻¹ has been suggested to increase the likelihood of an ergogenic benefit [11, 28, 41], therefore a time interval comprising those times where the lower limit of the 95% confidence interval (CI) for delta blood [HCO₃⁻] exceeded this threshold was estimated.

The longitudinal data of each individual session were also summarised as follows: i) A smooth curve was fitted using a cubic spline function with B-spline bases and one knot [42], and the curve used to calculate: a) peak (iPeak; the maximum of the predicted values (minimum for blood [Cl⁻] (iMin)), as ascertained for each participant during each individual session); b) change from baseline to iPeak or iMin (iDelta); and c) time to iPeak or iMin measured from the completion of ingestion; ii) Area under the curve (AUC) for each session was determined using the trapezoidal method on the raw data. These four summary measures were compared using LMMs with treatment as a fixed effect and participant as a random effect. The same LMM was used to compare palatability, urine volume excreted and urinary [citrate].

Gastrointestinal symptoms data displayed minimal variability, and was summarised as: a) sum of the rating of all symptoms at each time point (range 0–40; 10 symptoms each ranging from 0–4); b) total session GI symptoms rating calculated as the sum of the rating of all symptoms at all time points (range 0–720; 18 time points, maximum of 40 at each time point); c) number of participants that reported each symptom at any time point. The sum of all symptoms (a) and the total session rating (b) were reported as mean and range. Stata v15 was used for all analyses. Results were considered statistically significant when $p \leq 0.05$.

## Results

### Pattern over time

A main effect for time was detected in that all four treatments produced an increase in blood pH, [HCO₃⁻], [Na⁺] and a decrease in blood [Cl⁻] from baseline (Figs 1A, 1C, 2A and 2C). A main effect for treatment was detected for blood pH, [HCO₃⁻], [Na⁺] and [Cl⁻] (Table 1), with the respective pairwise comparisons of this effect displayed in S1 Table. No time by treatment interaction effects were detected for these blood variables (Figs 1B, 1D, 2B and 2D). There was a higher blood pH with the 15 min ingestion period compared with the 30, 45 and 60 min ingestion periods (Table 1). A higher blood [HCO₃⁻] was detected for the 15 and 45 min ingestion periods when compared with the 60 min ingestion period (Table 1). There was a lower blood [Na⁺] with the 30 min ingestion period when compared to the 60 min ingestion period,

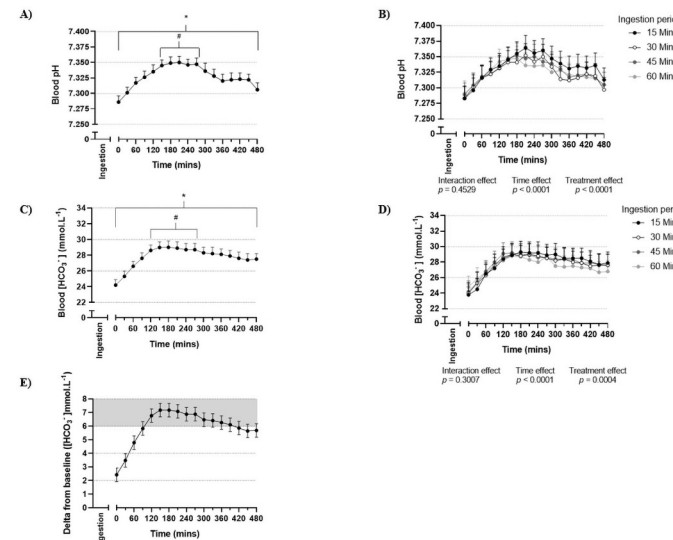

**Fig 1. Blood pH and blood [HCO$_3^-$] after 500 mg.kg$^{-1}$ BM sodium citrate ingestion.** (A) Blood pH irrespective of ingestion period ($n$ = 16 participants, 4 sessions per participant, 64 total observations). (B) Blood pH following the completion of ingestion of sodium citrate over 15, 30, 45 or 60 min ($n$ = 16 participants per ingestion period). (C) Blood [HCO$_3^-$] (mmol.L$^{-1}$) irrespective of ingestion period ($n$ = 16 participants, 4 sessions per participant, 64 total observations). (D) Blood [HCO$_3^-$] (mmol.L$^{-1}$) following the completion of ingestion of sodium citrate over 15, 30, 45 or 60 min ($n$ = 16 participants per ingestion period). (E) Delta blood [HCO$_3^-$] (mmol.L$^{-1}$) irrespective of ingestion period ($n$ = 16 participants, 4 sessions per participant, 64 total observations). Values are mean and 95% confidence intervals. Zero (0) value on the x-axis corresponds to the completion of sodium citrate ingestion. Grey (shaded) area in (E) indicates 6 mmol.L$^{-1}$ above baseline. $^*$ elevated ($p < 0.05$) compared to baseline. $^\#$ time interval where values are above baseline ($p < 0.05$) and not below gPeak (the maximum mean value, as ascertained from cumulative group data; $p > 0.05$).

and a lower blood [Cl$^-$] with the 15 min ingestion period when compared to the 60 min ingestion period. Ingestion period (treatment) altered the pattern over time for plasma [citrate] (Fig 2E, $p$ = 0.0324, time by treatment interaction), with the 15 min ingestion period associated with greater plasma [citrate] than other ingestion periods from 180 to 240 min post-ingestion.

## Baseline, peak and time to peak

No differences were detected between treatments for baseline values prior to sodium citrate ingestion for any blood variable (Tables 2 and 3). The overall time to gPeak was 210 min for blood pH (Fig 1A), 180 min for blood [HCO$_3^-$] (Fig 1D), 120 min for blood [Na$^+$] (Fig 2A), and 270 min for gMin for blood [Cl$^-$] (Fig 2D). The time interval where mean values were not significantly different from gPeak was 150–270 min for blood pH, and 120–270 min for blood [HCO$_3^-$]. The time interval where blood [HCO$_3^-$] significantly exceeded 6 mmol.L$^{-1}$ above baseline was 120–270 min (Fig 1E). No differences by treatment were detected for iPeak or iDelta for any variable (Tables 2 and 3, S2 and S3 Tables). Time to iPeak for blood pH occurred earlier for the 60 min ingestion period compared to the 15 and 45 min ingestion periods (Table 2, for pairwise comparisons see S2 Table). Time to iPeak for blood [HCO$_3^-$] occurred earlier for the 45 min ingestion period compared to the 15 min ingestion period (Table 2 and S2 Table).

## Gastrointestinal symptoms and palatability

The total session mean GI symptoms ratings (scale 0–720) ranged from 9.8 (60 min ingestion period) to 11.6 (15 min ingestion period) (Table 4). Mean GI symptoms (rating) did not

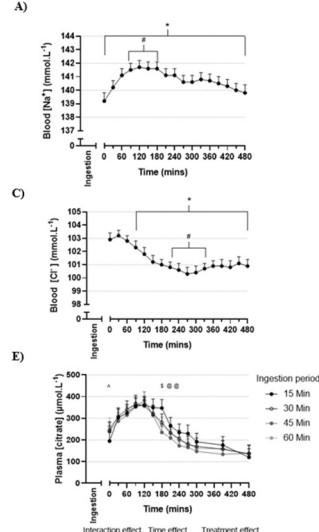
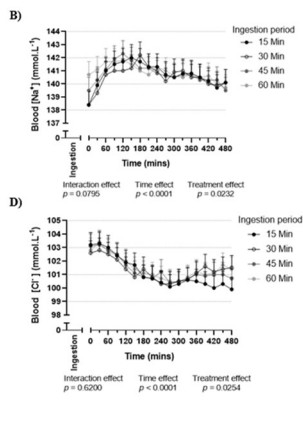

**Fig 2. Blood [Na⁺], blood [Cl⁻] and plasma [citrate] after 500 mg.kg⁻¹ BM sodium citrate ingestion.** (A) Blood [Na⁺] (mmol.L⁻¹) irrespective of ingestion period ($n = 16$ participants, 4 sessions per participant, 64 total observations). (B) Blood [Na⁺] (mmol.L⁻¹) following the completion of ingestion of sodium citrate over 15, 30, 45 or 60 min ($n = 16$ participants per ingestion period). (C) Blood [Cl⁻] (mmol.L⁻¹) irrespective of ingestion period ($n = 16$ participants, 4 sessions per participant, 64 total observations). (D) Blood [Cl⁻] (mmol.L⁻¹) following the completion of ingestion of sodium citrate over 15, 30, 45 or 60 min ($n = 16$ participants per ingestion period). (E) Plasma [citrate] (μmol.L⁻¹) following the completion of ingestion of sodium citrate over 15, 30, 45 or 60 min ($n = 16$ participants per ingestion period). Values are mean and 95% confidence intervals. Zero (0) value on the x-axis corresponds to the completion of sodium citrate ingestion. * elevated ($p < 0.05$) compared to baseline. # time interval where values are above baseline ($p < 0.05$) and not below gPeak (the maximum mean value, as ascertained from cumulative group data; $p > 0.05$). $ difference ($p < 0.05$) between 15 min all other ingestion periods. @ difference ($p < 0.05$) between 15 min and 45 min ingestion period. ^ difference ($p < 0.05$) between 15 min and 60 min ingestion period. gPeak for blood [Na⁺] occurred 120 min post-ingestion (A), gMin occurred 270 min post-ingestion for blood [Cl⁻] (C).

exceed 1.2 out of a maximum of 40 at any given measurement time (Fig 3A). The most frequently reported symptoms (irrespective of severity) were sickness and nausea, reported by between four and six of 16 participants per session (Fig 3C). No differences were detected between treatments for palatability (score) (Table 4 and S4 Table).

**Table 1. Mean (95% confidence intervals) of the treatment effect for post-ingestion session values for blood variables (pH, [HCO₃⁻], [Na⁺] and [Cl⁻]) following ingestion of 500 mg.kg⁻¹ BM sodium citrate ($n = 16$ participants, 18 observations per participant per treatment).**

|  | 15 min ingestion period | 30 min ingestion period | 45 min ingestion period | 60 min ingestion period | Treatment effect ($p$ value) |
|---|---|---|---|---|---|
| Blood pH | 7.330 (7.325, 7.336) | 7.318 (7.313, 7.324) [a] | 7.321 (7.315, 7.327) [a] | 7.321 (7.315, 7.326) [a] | < 0.0001 |
| Blood [HCO₃⁻] (mmol.L⁻¹) | 27.7 (27.3, 28.0) | 27.4 (27.0, 27.8) | 27.4 (27.0, 27.8) | 27.1 (26.7, 27.6) [a d] | 0.0004 |
| Blood [Na⁺] (mmol.L⁻¹) | 140.6 (140.3, 140.9) | 140.3 (140.0, 140.6) [c] | 140.6 (140.3, 140.8) | 140.7 (140.5, 141.0) | 0.0232 |
| Blood [Cl⁻] (mmol.L⁻¹) | 101.3 (101.0, 101.5) | 101.4 (101.1, 101.7) | 101.4 (101.2, 101.6) | 101.6 (101.4, 101.8) [b] | 0.0254 |

Estimates obtained under a linear mixed model including participants as a random effect and fixed effects: treatment, time (categorical), time by treatment interaction.

Estimates are presented only for blood variables with a non-significant interaction effect.

[a] 15-min ingestion period greater than compared ingestion period.

[b] 15-min ingestion period lower than compared ingestion period.

[c] 60-min ingestion period greater than compared ingestion period.

[d] 45-min ingestion period greater than compared ingestion period.

**Table 2. Estimates of curve characteristics for blood pH and blood bicarbonate concentration ([HCO$_3^-$]) following ingestion of 500 mg.kg$^{-1}$ BM sodium citrate over 15, 30, 45 or 60 min (*n* = 16 participants per treatment).**

| | 15 min ingestion period | 30 min ingestion period | 45 min ingestion period | 60 min ingestion period |
|---|---|---|---|---|
| Blood pH | | | | |
| Baseline ^ | 7.256 (7.230, 7.282) | 7.248 (7.222, 7.273) | 7.234 (7.208, 7.259) | 7.243 (7.218, 7.269) |
| iPeak ^ † | 7.357 (7.348, 7.366) | 7.352 (7.343, 7.361) | 7.361 (7.352, 7.371) | 7.351 (7.342, 7.360) |
| iDelta ^ † | 0.112 (0.102, 0.121) | 0.107 (0.098, 0.116) | 0.116 (0.107, 0.126) | 0.106 (0.097, 0.115) |
| Time to iPeak (min) ^ † | 240 (201, 279) [a] | 225 (186, 264) | 241 (203, 281) [a] | 167 (128, 206) |
| Area under the curve ^ | 3521 (3513, 3530) | 3516 (3507, 3524) | 3517 (3509, 3526) | 3517 (3509, 3525) |
| Time interval (min)# | 180–270 | 150–270 | 150–300 | 150–270 |
| Blood bicarbonate concentration ([HCO$_3^-$]) | | | | |
| Baseline (mmol.L$^{-1}$) ^ | 22.0 (20.8, 23.3) | 22.1 (20.9, 23.3) | 21.5 (20.3, 22.7) | 21.5 (20.3, 22.7) |
| iPeak (mmol.L$^{-1}$) ^ † | 29.6 (28.6, 30.5) | 29.3 (28.3, 30.2) | 29.7 (28.7, 30.6) | 29.2 (28.3, 30.2) |
| iDelta (mmol.L$^{-1}$) ^ † | 7.8 (6.8, 8.7) | 7.5 (6.5, 8.4) | 7.9 (7.0, 8.9) | 7.4 (6.5, 8.4) |
| Time to iPeak (min) ^ † | 251 (203, 300) | 242 (194, 290) | 182 (134, 230) [b] | 231 (182, 279) |
| Area under the curve ^ | 13504 (12861, 14146) | 13339 (12696, 13981) | 13386 (12744, 14028) | 13238 (12595, 13880) |
| Time interval (min) # | 120–390 | 120–330 | 120–360 | 120–270 |

^ mean (95% confidence interval), estimated under a linear mixed model (LMM) including treatment as fixed effect and participant as random effect.

† calculated from a smoothed curve for each participant during each individual session.

# time points where values were greater than baseline ($p < 0.05$) and not below gPeak ($p > 0.05$), estimated under a LMM including treatment, time (categorical) and their interaction. iPeak (the maximum value from each individual session); iDelta (change from baseline to iPeak); Time to iPeak (from completion of ingestion to iPeak).

[a] different ($p < 0.05$) from 60 min ingestion period.

[b] different ($p < 0.05$) from 15 min ingestion period.

## Urinary excretion

Mean pre-testing total urine volume was 1298 mL (95% CI: 889, 1706). There were no differences in total urine volume across treatments (Table 4), or between pre-testing urine collection and any of the sodium citrate treatments. The (mean, 95% CI) pre-testing excreted urinary citrate mass was 258.4 mg (117.2, 399.6). There was an increase in the amount of citrate excreted with all four sodium citrate treatments, with mean delta values ranging between 362.9 and 465.8 mg (Table 4).

## Discussion

The primary aim of this investigation was to compare the effect of varying sodium citrate ingestion periods (15, 30, 45 and 60 min) on blood alkalosis and GI symptoms. A key finding of this investigation was that ingestion of 500 mg.kg$^{-1}$ BM sodium citrate in gelatine capsules across a 60 min period was associated with a lower level of blood alkalosis (both blood pH and [HCO$_3^-$]) when compared to shorter ingestion periods. Blood alkalosis peaked at 180 (blood [HCO$_3^-$]) and 210 (blood pH) min after ingestion, and remained elevated above baseline for at least 480 min. Additionally, alkalosis (combining blood [HCO$_3^-$] and pH results) was both above baseline and not below the peak from 150–270 min after ingestion. The sodium citrate ingestion period did not impact palatability, and the mean reported GI symptoms were minor for all ingestion protocols at all times.

**Table 3. Estimates of curve characteristics for blood sodium ($[Na^+]$), blood chloride ($[Cl^-]$) and plasma citrate ([citrate]) following ingestion of 500 mg.kg$^{-1}$ BM sodium citrate over a 15, 30, 45 or 60 min period (*n* = 16 participants per treatment).**

| | 15 min ingestion period | 30 min ingestion period | 45 min ingestion period | 60 min ingestion period |
|---|---|---|---|---|
| Blood sodium concentration ($[Na^+]$) | | | | |
| Baseline (mmol.L$^{-1}$) ^ | 137.9 (136.8, 139.1) | 137.0 (135.8, 138.2) | 137.3 (136.1, 138.4) | 137.9 (136.8, 139.1) |
| iPeak (mmol.L$^{-1}$) ^ † | 142.2 (141.6, 142.9) | 142.3 (141.6, 143.0) | 142.6 (142.0, 143.3) | 142.2 (141.5, 142.8) |
| iDelta (mmol.L$^{-1}$) ^ † | 4.7 (4.1, 5.4) | 4.8 (4.1, 5.4) | 5.1 (4.4, 5.7) | 4.6 (4.0, 5.3) |
| Time to iPeak (min) ^ † | 210 (151, 269) | 193 (134, 252) | 180 (121, 239) | 139 (80, 198) |
| Area under the curve ^ | 67607 (67244, 67971) | 67493 (67129, 67856) | 67603 (67239, 67966) | 67662 (67298, 68025) |
| Time interval (min) # | 60–240 | 150–210 | 90–150 | 30–270 |
| Blood chloride concentration ($[Cl^-]$) | | | | |
| Baseline (mmol.L$^{-1}$) ^ | 103.0 (102.0, 104.0) | 102.9 (101.8, 103.9) | 102.4 (101.3, 103.4) | 103.4 (102.4, 104.5) |
| iPeak (mmol.L$^{-1}$) ^ † | 99.4 (98.7, 100.0) | 100.2 (99.6, 100.9) | 100.2 (99.5, 100.9) | 99.8 (99.2, 100.5) |
| iDelta (mmol.L$^{-1}$) ^ † | -3.6 (4.2, -2.9) | -2.7 (-3.3, -2.0) | -2.7 (-3.4, -2.0) | -3.1 (-3.8, -2.4) |
| Time to iPeak (min) ^ † | 394 (338, 449) | 296 (241, 352) [a] | 308 (252, 363) [a] | 289 (233, 344) [a] |
| Area under the curve ^ | 48546 (48195, 48896) | 48614 (48263, 48964) | 48632 (48282, 48983) | 48709 (48359, 49060) |
| Time interval (min) # | 180–480 | 150–420 | 150–480 | 150–480 |
| Plasma citrate concentration ([citrate]) | | | | |
| Baseline (µmol.L$^{-1}$) ^ | 141.8 (121.6, 162.0) | 131.3 (110.2, 152.4) | 130.5 (110.2, 150.7) | 129.2 (109.0, 149.4) |
| iPeak (µmol.L$^{-1}$) ^ † | 426.5 (386.9, 466.1) | 419.0 (378.2, 459.8) | 408.6 (369.2, 448.0) | 432.9 (393.5, 472.3) |
| iDelta (µmol.L$^{-1}$) ^ † | 293.4 (253.9, 333.0) | 285.9 (245.1, 326.7) | 275.5 (236.1, 314.9) | 299.8 (260.4, 339.2) |
| Time to iPeak (min) ^ † | 122 (100, 144) | 105 (83, 127) | 105 (83, 127) | 99 (77, 122) |
| Area under the curve ^ | 118653 (106058, 131249) | 111434 (98839, 124030) | 103743 (91148, 116339) | 111632 (98945, 124319) |
| Time interval (min) # | 60–180 | 60–120 | 90–120 | 90–120 |

^ mean (95% confidence interval), estimated under a linear mixed model (LMM) including treatment as fixed effect and participant as random effect.

† calculated from a smoothed curve for each participant during each individual session.

# time points where values were greater than baseline ($p < 0.05$) and not below gPeak ($p > 0.05$), estimated under a LMM including treatment, time (categorical) and their interaction. iPeak (the maximum value from each individual session); iDelta (change from baseline to iPeak); Time to iPeak (from completion of ingestion to iPeak). A different ($p < 0.05$) from 15 min ingestion period.

## Induced blood alkalosis

In the current investigation, the 30, 45 and 60 min ingestion periods were associated with a small, but significantly lower blood pH (~0.01 pH or 1 nmol.L$^{-1}$) when compared to the 15 min ingestion period, despite the identical sodium citrate dose and mode in each treatment. The 60 min ingestion period was associated with a mean session blood $[HCO_3^-]$ of 27.1 mmol.L$^{-1}$, compared to 27.7 mmol.L$^{-1}$ for the 15 min ingestion period. While statistically significant, this mean difference of 0.6 mmol.L$^{-1}$ is similar in magnitude to that of the width of the 95% confidence intervals for each treatment. Therefore, although statistical differences for blood alkalosis were detected when comparing treatments in some analyses, the small absolute size of these differences may suggest limited clinical differences with regards to blood alkalosis responses across treatments. Blood $[HCO_3^-]$ did not differ between the 15, 30 and 45 min ingestion periods, but blood pH was slightly lower in the latter two treatments compared with the 15 min ingestion period. Overall, it can therefore be concluded that ingesting sodium citrate over a 15, 30 or 45 min period may be associated with a similar or greater total blood alkalosis response than for a 60 min ingestion period. When considering both blood pH and $[HCO_3^-]$ as components of blood alkalosis, there is little difference between the 15, 30 and 45

**Table 4. Gastrointestinal symptoms (rating, _n_ = 16 participants), palatability (score, _n_ = 16 participants) and urinary excretion values (_n_ = 8 participants) following ingestion of 500 mg.kg$^{-1}$ BM sodium citrate over 15, 30, 45 or 60 min.**

| Additional measures | 15 min ingestion period | 30 min ingestion period | 45 min ingestion period | 60 min ingestion period |
|---|---|---|---|---|
| Gastrointestinal symptoms | | | | |
| Total session gastrointestinal symptoms rating (mean, range) | 11.6 (0–66) | 9.9 (0–37) | 10.4 (0–72) | 9.8 (0–50) |
| Palatability | | | | |
| Palatability score (mean, 95% confidence intervals) | 5.3 (4.4–6.1) | 5.9 (5.0–6.7) | 5.4 (4.5–6.2) | 5.5 (4.7–6.3) |
| Urinary excretion | | | | |
| Urinary [citrate] (µmol.L$^{-1}$) | 3360.6 (2695.8–4025.3) | 3223.4 (2558.6–3888.1) | 3184.0 (2519.2–3848.7) | 3305.8 (3641.0–3970.5) |
| Total excreted citrate (mg) | 660.3 (518.0–802.7) | 724.2 (581.9–866.5) | 621.3 (479.0–763.6) | 687.9 (545.6–830.2) |
| Delta excreted citrate (total excreted citrate minus pre-testing citrate (mg)) | 401.9 (222.4–581.5) | 465.8 (286.3–645.4) | 362.9 (183.4–542.5) | 429.5 (250.0–609.1) |
| Total urine excreted (mL) | 1113.8 (703.9–1523.8) | 1302.0 (892.0–1712.0) | 1254.5 (844.6–1664.5) | 1247.4 (837.4–1657.3) |

For total session gastrointestinal symptoms rating, the maximum possible value was a rating of 720 (maximum of 40 at each time point, across 18 time points). For palatability score, the maximum possible value was a score of 9.

min ingestion periods, suggesting that any of these three protocols could be recommended on the basis of the findings from this study.

Blood pH and [HCO$_3^-$] were elevated above baseline immediately post-ingestion and for the entire 480 min post-ingestion period. Prior investigations had undertaken post-ingestion observation periods with a duration of up to 240 min [4, 16], therefore the complete time interval where induced blood alkalosis was significantly elevated (beyond 240 min post-ingestion) was unknown. The time to iPeak for blood pH and/or [HCO$_3^-$] was significantly earlier when supplementation occurred over a longer period (i.e. 45 or 60 min). This is likely explained by

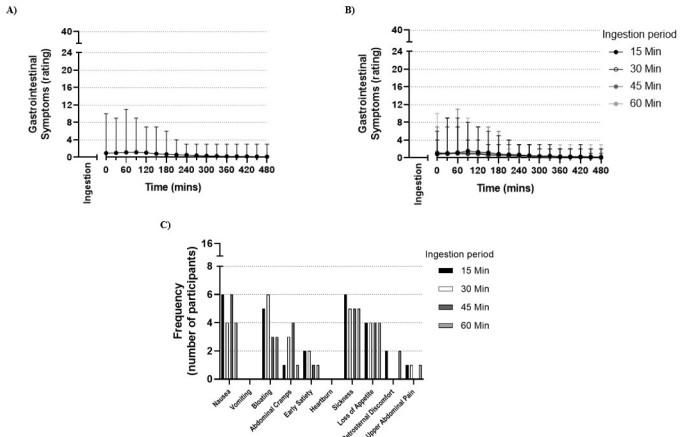

**Fig 3. Gastrointestinal symptoms after 500 mg.kg$^{-1}$ BM sodium citrate ingestion.** (A) Mean and range of gastrointestinal symptoms (rating) following completion of ingestion of sodium citrate, with a maximum possible rating of 40 at each time-point, irrespective of ingestion period (_n_ = 16 participants, 4 sessions per participant, 64 total observations). (B) Mean and range of gastrointestinal symptoms (rating) following completion of ingestion of sodium citrate over 15, 30, 45 or 60 min, with a maximum possible rating of 40 at each time-point (_n_ = 16 participants per ingestion period). (C) Frequency of each gastrointestinal symptom reported by all participants after ingestion of sodium citrate over 15, 30, 45 or 60 min (_n_ = 16 participants). For both (A) and (B), zero (0) value on the x-axis corresponds to the completion of sodium citrate ingestion.

the earlier relative timing of the first exposure to the supplement in the longer periods, as the 60 min ingestion period received the first sub-dose of the supplement 45 min earlier than the 15 min ingestion period and so on. This effect may be considered similar to that previously reported within the sodium bicarbonate literature, when comparing gelatine capsules to delayed-release capsules. Delayed-release or enteric-coated capsules have been reported to delay the occurrence of peak blood alkalosis when compared to gelatine capsules (by approx. 20–24 min [43, 44]) in a similar fashion to that seen when comparing the 15 and 60 min ingestion periods in the current investigation (by approx. 20 min). Peak blood alkalosis was observed at approximately similar times when assessed via gPeak (180–210 min post-ingestion) or iPeak (167–251 min post-ingestion), both consistent with findings from a prior investigation which implemented the same sodium citrate dose and mode of ingestion [16].

An increase in blood $[HCO_3^-]$ exceeding 6 mmol.L$^{-1}$ may increase the likelihood of improved exercise performance following ingestion of buffering agents [11, 28]. In the current investigation, a 6 mmol.L$^{-1}$ increase equated to a time period from 120–270 min post-ingestion, irrespective of the ingestion period. It must be acknowledged that no study has established the threshold upon which the blood $[HCO_3^-]$ must increase to improve high-intensity exercise performance following ingestion of sodium citrate. As such, a time interval within which blood $[HCO_3^-]$ is elevated above baseline, but not below the peak may be of relevance. In the current investigation, this was observed from 120–270 min post-ingestion, matching the time interval for the 6 mmol.L$^{-1}$ threshold. Whether there is an ideal time to commence exercise within this 150 min time interval has yet to be determined. Experimental results from the sodium bicarbonate literature have suggested that commencing exercise in correspondence with individual peak blood alkalosis is associated with greater performance benefit than using a post-ingestion timing that is standardised across all participants or athletes [28]. However, the poor intra-individual reliability of time to peak blood $[HCO_3^-]$ reported in a recent sodium bicarbonate investigation [45] questions the validity of supplementation strategies based on individualised time to peak blood alkalosis. Further work is required to ascertain the intra-individual reliability of blood alkalosis responses to sodium citrate supplementation. Future research should also quantify intramuscular acid-base responses to sodium citrate supplementation, to complement the established blood responses, and to build on the findings of the sole sodium citrate investigation that has previously measured muscle pH [46].

Both blood $[Na^+]$ and plasma [citrate] peaked at similar times to those reported in prior investigations where participants ingested 500 mg.kg$^{-1}$ BM sodium citrate [17, 46]. Sodium citrate supplementation has been suggested to induce blood alkalosis following an increased SID resulting from relative differences in the rate of entry and removal of citrate and sodium ions from the circulation [31, 47]. Subsequent increases in $H^+$ excretion and/or decreases in $HCO_3^-$ excretion from the kidneys are expected [30], which restore electrical equilibrium by increasing blood pH and $[HCO_3^-]$ (blood alkalosis). While both plasma [citrate] and blood $[Na^+]$ were elevated immediately post-ingestion, the relative change in concentration was of greater magnitude for blood $[Na^+]$ than for plasma [citrate]. When converting to standardised units ($\mu$mol.L$^{-1}$), the change from baseline to immediately post-ingestion was approximately 2000 $\mu$mol.L$^{-1}$ for blood $[Na^+]$ (from 137.5 to 139.5 mmol.L$^{-1}$, one positive charge per sodium ion) and approximately 300 $\mu$mol.L$^{-1}$ for plasma [citrate] (from 132 to 230 $\mu$mol.L$^{-1}$, three negative charges per citrate ion). This greater increase in blood $[Na^+]$ compared to plasma [citrate] likely represents a change in SID, to be followed by induced blood alkalosis. The 90 min delay between ingestion and a significant decline in blood $[Cl^-]$ may indicate that the SID response to sodium citrate supplementation is not highly dependent on blood $[Cl^-]$. Assessment of all strong ions (potassium, magnesium, calcium and lactate) is required to establish a more complete understanding of the SID response to sodium citrate supplementation.

## Handling of sodium, chloride and citrate ions

The similarity across treatments for iPeak, iMin and iDelta for blood $[Na^+]$, $[Cl^-]$ and plasma [citrate] may represent a lack of an effect of ingestion period on the magnitude of change in SID, although future monitoring of changes in all strong ions is needed to confirm this. With some variability according to ingestion period (significant treatment by time interaction effect), plasma [citrate] returned to baseline concentrations approximately 240–300 min post-ingestion. This occurred one to two hours later than that seen in prior investigations of plasma [citrate] [48, 49]. These differences are most likely due to the greater citrate load ingested in the current investigation (500 mg.kg$^{-1}$ BM sodium citrate dose vs 40 mEq. potassium citrate (approximately 185 mg.kg$^{-1}$ BM dose)), and/or possibly due to the differing ingestion modes implemented across studies (gelatine capsules in the current investigation, tablets or solution in prior investigations [48, 49]). Elevated blood $[Na^+]$ persisted for the entire 480 min post-ingestion observation period, an outcome not previously assessed in sodium citrate or similar literature. The differences in blood $[Na^+]$ and plasma [citrate] may relate to the differing pathway(s) through which these ions are absorbed and removed from the circulation. The earlier return to baseline concentrations for plasma [citrate] compared to blood $[Na^+]$, may indicate that the rate of removal exceeded the rate of entry at an earlier point for citrate ions than for sodium ions.

Urinary citrate excretion (mg) was increased when ingesting sodium citrate for all ingestion periods compared to no sodium citrate ingestion (i.e. urine collected prior to the first session) (all $p < 0.001$). Urinary citrate excretion (mg) increased above pre-testing concentrations by 363 to 466 mg, with no difference between ingestion periods (all $p > 0.05$). No prior investigation has assessed the impact of sodium citrate ingestion on urinary citrate excretion, although potassium citrate supplementation is associated with increased urinary citrate excretion [49]. This, together with the ingested sodium citrate dose (500 mg.kg$^{-1}$ BM), may facilitate greater understanding of citrate metabolism. The mass of sodium citrate ingested per session was approximately 34 g, of which ~25 g was citrate (assuming negligible amounts of citrate in the co-ingested meal). Over the 480 min post-ingestion period, only ~0.4 g of the ingested citrate load (~2% of total ingested) was excreted into the urine. Given that citrate absorption in the GI system has been reported to be nearly complete [48], it is unlikely that a substantial portion of the ingested citrate load passed directly into the faeces without entering the circulation. Plasma [citrate] returning to resting concentrations 240–300 min post-ingestion indicates that the remaining 98% of the ingested citrate load must be accounted for by removal from the circulation via pathways other than urinary excretion, potentially involving the kidneys, liver and skeletal muscle [46, 50].

## Gastrointestinal symptoms

Sodium citrate ingestion was associated with very minor GI symptoms for all ingestion periods. When considering the timing of GI symptoms, participants reported the highest ratings approximately 60–90 min post-ingestion, consistent with prior investigations [4, 16]. Gastrointestinal symptoms were minor at each time point, with a decline below a mean rating of 1.0 out of 40 from 150 min post-ingestion and onwards. The minor GI symptoms reported by participants was expected given our ingestion protocol, including the sodium citrate dose of 500 mg.kg$^{-1}$ BM [4], capsules rather than solution as the ingestion mode [16], a fluid volume below 800 mL [51], and a co-ingested meal [52]. These findings are important for athletes, suggesting they can ingest sodium citrate at times in line with peak blood alkalosis (150–270 min) before exercise with minimal GI symptoms.

## Palatability

No difference was detected when comparing the sodium citrate ingestion periods for palatability ($p > 0.05$). Palatability (across treatments) ranged from 5.3 to 5.9, which corresponds to a rating of between 'neither like nor dislike' and 'like slightly' [40]. A low palatability, especially for foods or fluids that are excessively salty, can be associated with avoidance of that particular food or fluid [53, 54]. From a practical perspective, higher palatability ratings may increase the extent to which a dietary supplementation strategy could be implemented in the context of high-intensity exercise. The broad similarity in palatability across treatments was to be expected, given the standardisation of the sodium citrate dose, ingestion mode, and co-ingested food and fluid. The moderate palatability ratings reported by participants of this investigation indicate that sodium citrate may be feasible for inclusion in training or competition routines.

## Limitations

There are some limitations to the research design. The absence of an exercise performance test in this study reduces the extent to which findings can be directly applied to athletes. However this was a necessary element within the design of the current investigation, to isolate the impact of sodium citrate ingestion on physiological responses and gastrointestinal symptoms over an extended period of time. Further, the recruitment of healthy participants rather than trained individuals/athletes may limit transferability of these findings to the nutritional practices of athletes. Future investigations should assess exercise performance in a well-trained population when ingesting sodium citrate according to the ingestion protocols used within the current investigation. Finally, the sample size used in this study was based on pragmatic considerations, due to a lack of available data from comparable investigations assessing the effect of varying sodium citrate ingestion periods. However, the final sample size of 16 participants in a cross-over design where four conditions were tested exceeds the sample sizes of the majority of prior sodium citrate investigations.

## Practical application statement

Based on the findings of the current investigation, it is recommended that sodium citrate be ingested across a 15–45 min period, with ingestion completed 150–270 min before the commencement of exercise, particularly for individuals competing in events where there may be performance benefit after induced blood alkalosis. Adherence to the recommended dose of 500 mg.kg$^{-1}$ BM, ingested in gelatine capsules alongside a small carbohydrate-rich meal is recommended.

## Novelty statement

This is the first investigation to monitor blood alkalosis for a period exceeding 240 min after ingestion of sodium citrate. No prior investigation has compared different sodium citrate ingestion periods. This investigation identifies a time interval where blood alkalosis is above baseline and also not below the peak (150–270 min post-ingestion), which provides additional information with regards to the optimum time to perform intense exercise after sodium citrate ingestion, for implementation by athletes. Monitoring of blood [Na$^+$], blood [Cl$^-$], plasma [citrate], and urinary excretion had not previously been conducted following sodium citrate ingestion, and may provide a greater understanding of the pathways through which sodium citrate induces blood alkalosis.

## Supporting information

**S1 Table. Pairwise comparisons (mean difference, 95% CI) of post-ingestion session values for blood variables (blood pH, blood [HCO$_3^-$], blood [Na$^+$] and blood [Cl$^-$]) following ingestion of 500 mg.kg$^{-1}$ BM sodium citrate over 15, 30, 45 or 60 min ($n$ = 16 participants, 18 observations per participant per treatment).**
(DOCX)

**S2 Table. Pairwise comparisons (mean difference, 95% CI) of curve characteristics for blood pH and blood bicarbonate concentration ([HCO$_3^-$]) following ingestion of 500 mg.kg$^{-1}$ BM sodium citrate over 15, 30, 45 or 60 min ($n$ = 16 participants, 18 observations per participant per treatment).**
(DOCX)

**S3 Table. Pairwise comparisons (mean difference, 95% CI) of curve characteristics for blood sodium concentration ([Na$^+$]), blood chloride concentration ([Cl$^-$]) and plasma citrate concentration ([citrate]) following ingestion of 500 mg.kg$^{-1}$ BM sodium citrate over 15, 30, 45 or 60 min ($n$ = 16 participants, 18 observations per participant per treatment).**
(DOCX)

**S4 Table. Pairwise comparisons (mean difference, 95% CI) of palatability ($n$ = 16 participants) and urinary excretion values ($n$ = 8 participants) following ingestion of 500 mg.kg$^{-1}$ BM sodium citrate over 15, 30, 45 or 60 min.**
(DOCX)

## Author Contributions

**Conceptualization:** Charles S. Urwin, Rodney J. Snow, Dominique Condo, Glenn D. Wadley, Amelia J. Carr.

**Data curation:** Charles S. Urwin, Liliana Orellana.

**Formal analysis:** Charles S. Urwin, Liliana Orellana.

**Project administration:** Charles S. Urwin.

**Supervision:** Rodney J. Snow, Dominique Condo, Glenn D. Wadley, Amelia J. Carr.

**Writing – original draft:** Charles S. Urwin, Rodney J. Snow, Liliana Orellana, Dominique Condo, Glenn D. Wadley, Amelia J. Carr.

**Writing – review & editing:** Charles S. Urwin, Rodney J. Snow, Liliana Orellana, Dominique Condo, Glenn D. Wadley, Amelia J. Carr.

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
