## [Decision Letter · Decision Letter 0]

23 Feb 2021

PONE-D-21-03700

Does varying the ingestion duration of sodium citrate influence blood alkalosis and gastrointestinal symptoms?

PLOS ONE

Dear Dr. Urwin,

Thank you for submitting your manuscript to PLOS ONE. After careful consideration, we feel that it has merit but does not fully meet PLOS ONE’s publication criteria as it currently stands. Therefore, we invite you to submit a revised version of the manuscript that addresses the points raised during the review process.

We look forward to receiving your revised manuscript.

Kind regards,

Lars McNaughton, PhD

Academic Editor

PLOS ONE

Journal Requirements:

Reviewers' comments:

Reviewer's Responses to Questions

**Comments to the Author**

1. Is the manuscript technically sound, and do the data support the conclusions?

Reviewer #1: Yes

Reviewer #2: Yes

2. Has the statistical analysis been performed appropriately and rigorously? 

Reviewer #1: Yes

Reviewer #2: Yes

3. Have the authors made all data underlying the findings in their manuscript fully available?

Reviewer #1: No

Reviewer #2: Yes

4. Is the manuscript presented in an intelligible fashion and written in standard English?

Reviewer #1: Yes

Reviewer #2: Yes

5. Review Comments to the Author

Reviewer #1: This study compared different ingestions durations of the same dose of sodium citrate on blood alkalosis, gastrointestinal symptoms and indicators of strong ion difference. The study is well performed and the manuscript well written, although I do have some minor comments and suggestions that I hope the authors will consider.

General

It is unclear why VO2peak was determined. Is the pharmacokinetic response following supplementation modified by VO2peak? Are these data necessary to the study?

I think it is important to include a section (or statements throughout) specifying limitations of the study, of which I believe the following should be included: a lack of a performance outcome to test whether these differences are actually meaningful.

The large window of opportunity for an ergogenic effect is one we recently saw with sodium bicarbonate (DOI: 10.1249/MSS.0000000000002313). Alongside the lack of statistical difference between peak bicarbonate and bicarbonate in the time 120-270 min post-ingestion, which we also showed, I believe these data question the necessity for the time to peak strategy. The authors might wish to consider some discussion on this topic. I believe their data is also relevant for more prolonged high-intensity exercise where increased buffering capacity might be useful over a longer period (e.g. cycling – see Dalle et al. DOI: 10.1016/j.jsams.2020.09.011). Again, some relevant discussion might be of interest based upon these results.

Specific

Line 41 – What about muscle acidosis? Is blood acidosis not somewhat a reflection of what is happening at the muscle level?

Line 96 – Please use metre as the SI unit for height

Line 125 – How much sodium is in Powerade? Could this have influenced absolute changes?

Line 133-136: Just a suggestion but a figure might be useful to visualise the different timings.

Lines 332-334: I wonder what the physiological relevance of such minor differences are? I would like to see more emphasis on the actual differences in addition to the statistics. These differences are just as likely to be within measurement error or biological variability.

Line 348: “This may be explained…” - I would say this is almost entirely explained by this.

Lines 365-367: I would urge some caution here as this is only one study and the 60 min comparison (vs TTP) is certainly on the lower end of the time spectrum for bicarbonate increases.

Lines 390-391: “indicating that ingestion duration did not impact the magnitude of change in SID.” – I am not sure you can state that since you did not actually measure SID. I suggest reformulating this section.

Reviewer #2: General comments

This is an interesting and potentially important study for the exogenous buffer literature. The study is well conducted and controlled. There is a vast amount of data representing a very high volume of work which has been conducted. There are important findings and useful practical recommendations for those choosing to use sodium citrate as an exogenous buffering agent. I do however have some concerns about the presentation of some parts of the manuscript which the authors need to address in order to make the information more accessible to the readership. I’ve tried to make the comments as useful as possible, so hopefully they’ll be helpful.

Specific comments

Introduction

Line 45: It would be useful to state why there is likely lower GIS with citrate.

Line 46: There is a very abrupt start to this paragraph and it would benefit from either a more introductory statement, or a better link from the previous paragraph.

Line 47: “identified that ergogenic benefit…” should have an “an” between that and ergogenic.

Line 49: move (> 100% … to after “exercise”.

Lines 54-55: state why this ingestion time has previously been suggested.

Line 58: The clarity of your terminology regarding the “duration of ingestion” is really confusing. The reader can easily confuse this with the pre-exercise ingestion period, so it is important for you to define exactly what you mean here. It took me a few attempts to read this section and fully understand it (this may well be a reflection on me of course, but it’s likely to also be the case for others).

Line 62: This feels a bit of a dead end. Where does it lead, where’s the “so what?”

Lines 63-67: See previous comment about clarity of this ingestion period information.

Line 78: The Heibel et al and Jones et al., papers might be better here.

Line 81: “persistence” is an interesting choice of word, but it probably needs a bit of clarification.

Line 83: “changed” might be better as “alterations to” or something similar.

Line 91: Check the grammar here.

Methods

Line 95: Why 16?

Line 96: height should always be described in m not cm.

Lines 104-105: you don’t need to state the units for height and weight, it’s obvious, and you’ve already done it.

Lines 119-120: “10:00 pm the night prior” is a little awkwardly written. Just state the at washout period between trials was… you don’t need to clarify this with two sets of terms.

Line123: Please state the size of the capsules used to allow for possible replication.

Line 127: Why was such a small meal chosen. How does this relate to likely practice in feeding prior to exercise?

Line 135: it is unclear if this was exactly the same meal prescribed again. Please clarify.

Lines 147-148: Whilst the radiometer’s validity and reliability are well established, can you provide some estimate or support for this assay’s TEM/CV or validity and reliability.

Line 151: add “the” between “and” and “volume”.

Lines 164-165: This is a statement, not a paragraph. It should also be placed last in this section, as it’s not an appropriate starting statement for a statistical analysis section.

Lines 166-170: how did you determine the most appropriate model to use? What were the criteria? It would also be best practice to provide an estimate of effect size for these analyses. For the LMM you could consider Cohen’s F squared.

Line 178: Don’t use personal pronouns in scientific writing. I know it’s become fashionable of late to do this, but it’s not good practice. Please do this for the other instances of this in the manuscript (we and our) are used on multiple occasions.

Line 190; Don’t start sentences with abbreviations. Check this throughout the manuscript.

Lines 194-195: “variables a and b..” this needs more clarity.

Results

This is the section that I had some serious trouble with. I think it needs a complete re-think, because I got completely lost whilst reading it, and given that I have good knowledge and experience of this type of study, that means that most other will also likely struggle to access the information. The main issue is that there is just so much data and making comparison between the trial conditions is really difficult to follow. This isn’t helped by the clarity of some of the figures, which need solid lines and markers, and they need some of the error bars removing so that the actual data is visible, because it just look far to untidy at present.

The key focus must surely be the comparison in the responses to the ingestion time periods, so I think you need to display all of the variable with that in mind, to allow the reader to make this visual comparisons for themselves. I was a little unsure if the figure titles were titles, or sub-sectioned paragraphs as they seemed to be presented in the middle of the results section, which was a little confusing.

One strategy for making the description of the results easier to read, it to avoid using the term “significant” as much as possible. That forces you to write about data, rather than the stats outcomes. Remember that the stats outcomes, are there as supporting evidence, they should not be the sole focus. Try to describe the response of the variables and that show improve this section.

Discussion

Generally this was a good section with some nice flow and useful information and interpretation.

Line 329: given the size of some of the error bars, certainly initially, I’m not convinced that all of the participants would calls their GIS as “minor”. So can you re-phrase this to reflect that.

Line 336: replace “levels” with “concentration” throughout the manuscript when describing a blood or urine parameter.

Line 338: we can assume this is significant, so you don’t need the p-value.

Line 346: this is awkwardly phrased.

Line 350: “dosage” is a poor term. Dose is better. It would be worth clarifying this type of effect, to that seen with delayed-release capsule ingestion as the response of more gradual absorption is similar.

Line 354: “our laboratory” is a bit pretentious. Is this really needed? Just reference the paper.

Lines 362-368: Nice section.

Double check you use of square parentheses throughout the manuscript, especially in sentences where you have them next to “blood”.

Lines 394-396: This may have been due to capsule size differences.

The discussion might be better with the subheading removed, and the material in each included in other paragraphs or elsewhere (just a consideration really).

Lines 431-439: This section reads like a results section, rather than a discussion.

Lines 441: can you really conclude that athletes can use this strategy, given that you have recruited active participants?

Figures and Tables

Tables: these are generally clear, but they would benefit from the removal of as many borders and lines as possible.

Figures: add line markers and remove some of the error bars for clarity. Don’t use letters to highlight significance as the figures with multiple graphs on are also labelled with letters.

Make it easier for the reader to make comparisons between trials.

6. PLOS authors have the option to publish the peer review history of their article (what does this mean?). If published, this will include your full peer review and any attached files.

Reviewer #1: No

Reviewer #2: **Yes: **S. Andy Sparks

---

## [Author Response · Author response to Decision Letter 0]

20 Apr 2021

Does varying the ingestion duration of sodium citrate influence blood alkalosis and gastrointestinal symptoms?

Please note: line and page numbers in this document align with those in the final ‘Manuscript’ document submitted as a part of this revision.

Reviewer 1 

This study compared different ingestions durations of the same dose of sodium citrate on blood alkalosis, gastrointestinal symptoms and indicators of strong ion difference. The study is well performed and the manuscript well written, although I do have some minor comments and suggestions that I hope the authors will consider.

Thank you for taking the time to review our manuscript. The authors have sought to address all comments both below and in the revised manuscript.

General

It is unclear why VO2peak was determined. Is the pharmacokinetic response following supplementation modified by VO2peak? Are these data necessary to the study?

Thank you for your comment. The VO2peak assessment was conducted to provide a measure of cardiovascular fitness, as a component of health, alongside screening for kidney disease or use of blood acidity regulating medication. While these data are not included as an outcome measure or inclusion criteria per-se, they provide an indicator that our participant group had a reasonable level of aerobic fitness, which corresponds to terms such as “healthy, habitually active, physically fit, physically active, active and habitually physically active”, per a review by De Pauw and colleagues (https://doi.org/10.1123/ijspp.8.2.111). We have now added further detail to the Methods section on lines 117-120 on page 6 of the revised manuscript.

“VO2peak was determined as previously described by Urwin et al. [16], using an incremental increase in cycling intensity to volitional fatigue. These respiratory data were used to categorise participants “healthy” or “physically fit” according to previously used nomenclature [32].”

I think it is important to include a section (or statements throughout) specifying limitations of the study, of which I believe the following should be included: a lack of a performance outcome to test whether these differences are actually meaningful.

Thank you for providing this suggestions. The authors have now added a sub-section to the end of the discussion section, titled ‘Limitations’, which can be found on lines 459-471 on page 26 of the revised manuscript. This sub-section addresses the absence of an exercise performance test, the assessment of healthy participants (rather than elite athletes) and factors related to determining an appropriate sample size.

“There are some limitations to the research design. The absence of an exercise performance test in this study reduces the extent to which findings can be directly applied to athletes. However this was a necessary element within the design of the current investigation, to isolate the impact of sodium citrate ingestion on physiological responses and gastrointestinal symptoms over an extended period of time. Further, the recruitment of healthy participants rather than trained individuals/athletes may limit transferability of these findings to the nutritional practices of athletes. Future investigations should assess exercise performance in a well-trained population when ingesting sodium citrate according to the ingestion protocols used within the current investigation. Finally, the sample size used in this study was based on pragmatic considerations, due to a lack of available data from comparable investigations assessing the effect of varying sodium citrate ingestion periods. However, the final sample size of 16 participants in a cross-over design where four conditions were tested exceeds the sample sizes of the majority of prior sodium citrate investigations.” 

The large window of opportunity for an ergogenic effect is one we recently saw with sodium bicarbonate (DOI: 10.1249/MSS.0000000000002313). Alongside the lack of statistical difference between peak bicarbonate and bicarbonate in the time 120-270 min post-ingestion, which we also showed, I believe these data question the necessity for the time to peak strategy. The authors might wish to consider some discussion on this topic. 

Thank you for highlighting this important discussion point. The authors have accepted this comment and added this discussion to lines 371-378 on page 22 of the revised manuscript.

“Experimental results from the sodium bicarbonate literature have suggested that commencing exercise in correspondence with individual peak blood alkalosis is associated with greater performance benefit than using a post-ingestion timing that is standardised across all participants or athletes [28]. However, the poor intra-individual reliability of time to peak blood [HCO3-] reported in a recent sodium bicarbonate investigation [45] questions the validity of supplementation strategies based on individualised time to peak blood alkalosis. Further work is required to ascertain the intra-individual reliability of blood alkalosis responses to sodium citrate supplementation.”

I believe their data is also relevant for more prolonged high-intensity exercise where increased buffering capacity might be useful over a longer period (e.g. cycling – see Dalle et al. DOI: 10.1016/j.jsams.2020.09.011). Again, some relevant discussion might be of interest based upon these results.

Thank you for identifying this point. In response to this comment, we have made changes to the Introduction, on lines 55-56 on page 3 of the revised manuscript. Given the absence of an exercise performance test in the current investigation, the authors have not adjusted parts of the Discussion related to the specific types of exercise/event to which these physiological results might be applied.

“Performance of an all-out or high-intensity exhaustive effort at the end of endurance exercise has also been reportedly improved by supplementation with sodium citrate [13, 14].”

Specific

Line 41 – What about muscle acidosis? Is blood acidosis not somewhat a reflection of what is happening at the muscle level? 

Thank you for providing this comment. The authors have considered this comment carefully, but have referred to ‘blood acidosis’ as there is currently a relative dearth of information pertaining to the intramuscular responses to sodium citrate supplementation. More direct assessment of the intramuscular environment following sodium citrate supplementation is needed to confirm this as the mechanism through which the supplement may affect an improved exercise performance. The authors have therefore added a comment to this effect on lines 378-381 on page 22 of the revised manuscript.

“Future research should also quantify intramuscular acid-base responses to sodium citrate supplementation, to complement the established blood responses, and to build on the findings of the sole sodium citrate investigation that has previously measured muscle pH [46].”

Line 96 – Please use metre as the SI unit for height.

Thank you for highlighting this. The authors have now adjusted the unit for height to metres rather than centimetres, as is reflected on line 108 on page 5 of the revised manuscript.

Line 125 – How much sodium is in Powerade? Could this have influenced absolute changes? 

Thank you for identifying this. Whilst the quantity of sodium in Powerade was included in the original manuscript, we have now modified the wording within the revised manuscript on lines 142-144 on page 7 of the Methods section, to present this information more clearly. 

“According to manufacturer packaging, the co-ingested meals (including Powerade) contained a total of approximately 2 g of sodium, representing 18% of the total ingested sodium (given that approximately 9 g sodium was ingested as sodium citrate).”

The contribution of the co-ingested meals to the total amount of ingested sodium per session is small relative to that of the sodium citrate supplementation. Given the molecular weight of sodium citrate (258.06 g.mol-1), we can estimate that our participants ingested approximately 9 g of sodium per session from sodium citrate (dictated by the mean body mass of our participants, 68.3 kg), considerably more than the 2 g of sodium from the co-ingested meals. The primary focus of this investigation was the comparison of ingestion periods, so the magnitude of the absolute change in blood sodium concentration has not been discussed at length. Further, the same small quantity of sodium was ingested within the Powerade and co-ingested meals across all treatments according to our dietary standardisation procedures, which therefore reduces the possibility of the meals leading to any changes in blood sodium concentration between treatments.

Line 133-136: Just a suggestion but a figure might be useful to visualise the different timings. 

Thank you for providing this suggestion. The authors have considered including a figure depicting the timeline of data collection, but have decided to retain the description in the text alone. This decision was made primarily due to the current number of figures already included in this manuscript (seven). Reviewer 2 has requested that we include additional detail in the originally submitted tables, which the authors have added in the supplementary tables (Tables S1, S2, S3, S4). 

Lines 332-334: I wonder what the physiological relevance of such minor differences are? I would like to see more emphasis on the actual differences in addition to the statistics. These differences are just as likely to be within measurement error or biological variability. 

Thank you for highlighting this consideration. The authors now have addressed this on lines 328-344 on pages 20-21 of the revised manuscript. Further, the authors have adapted the results section to highlight clinical/physiological differences rather than simply p-values, per Reviewer 2 request.

“In the current investigation, the 30, 45 and 60 min ingestion periods were associated with a small, but significantly lower blood pH (~0.01 pH or 1 nmol.L-1) when compared to the 15 min ingestion period, despite the identical sodium citrate dose and mode in each treatment. The 60 min ingestion period was associated with a mean session blood [HCO3-] of 27.1 mmol.L-1, compared to 27.7 mmol.L-1 for the 15 min ingestion period. While statistically significant, this mean difference of 0.6 mmol.L-1 is similar in magnitude to that of the width of the 95% confidence intervals for each treatment. Therefore, although statistical differences for blood alkalosis were detected when comparing treatments in some analyses, the small absolute size of these differences may suggest limited clinical differences with regards to blood alkalosis responses across treatments. Blood [HCO3-] did not differ between the 15, 30 and 45 min ingestion periods, but blood pH was slightly lower in the latter two treatments compared with the 15 min ingestion period. Overall, it can therefore be concluded that ingesting sodium citrate over a 15, 30 or 45 min period may be associated with a similar or greater total blood alkalosis response than for a 60 min ingestion period. When considering both blood pH and [HCO3-] as components of blood alkalosis, there is little difference between the 15, 30 and 45 min ingestion periods, suggesting that any of these three protocols could be recommended on the basis of the findings from this study.” 

Line 348: “This may be explained…” - I would say this is almost entirely explained by this. 

The authors have adjusted the phrasing on lines 350-351 of page 21 of the revised manuscript.

 “This is likely explained by…”

Lines 365-367: I would urge some caution here as this is only one study and the 60 min comparison (vs TTP) is certainly on the lower end of the time spectrum for bicarbonate increases. 

Thank you for identifying this. The authors have made adjustments to this section on lines 371-378 on page 22 of the revised manuscript, as also documented above in response to your earlier comment.

“Experimental results from the sodium bicarbonate literature have suggested that commencing exercise in correspondence with individual peak blood alkalosis is associated with greater performance benefit than using a post-ingestion timing that is standardised across all participants or athletes [28]. However, the poor intra-individual reliability of time to peak blood [HCO3-] reported in a recent sodium bicarbonate investigation [45] questions the validity of supplementation strategies based on individualised time to peak blood alkalosis. Further work is required to ascertain the intra-individual reliability of blood alkalosis responses to sodium citrate supplementation.”

Lines 390-391: “indicating that ingestion duration did not impact the magnitude of change in SID.” – I am not sure you can state that since you did not actually measure SID. I suggest reformulating this section.

Thank you for highlighting this point. The authors have adjusted the phrasing on lines 401-403 on page 23 of the revised manuscript to reflect the results of the current investigation only. 

“The similarity across treatments for iPeak, iMin and iDelta for blood [Na+], [Cl-] and plasma [citrate] may represent a lack of an effect of ingestion period on the magnitude of change in SID, although future monitoring of changes in all strong ions is needed to confirm this.” 

Reviewer 2

General

This is an interesting and potentially important study for the exogenous buffer literature. The study is well conducted and controlled. There is a vast amount of data representing a very high volume of work which has been conducted. There are important findings and useful practical recommendations for those choosing to use sodium citrate as an exogenous buffering agent. I do however have some concerns about the presentation of some parts of the manuscript which the authors need to address in order to make the information more accessible to the readership. I’ve tried to make the comments as useful as possible, so hopefully they’ll be helpful.

Thank you for taking the time to review our manuscript and provide these thoughtful comments. The authors have sought to address each of these comments in detail. 

Introduction

Line 45: It would be useful to state why there is likely lower GIS with citrate. 

Thank you for identifying this key detail. There is currently little experimental evidence on which to base an explanation for the potential differences between sodium citrate and sodium bicarbonate in terms of gastrointestinal (GI) symptoms. A recent investigation by Peacock et al. (2021) did however compare the two supplements for their respective GI symptoms responses, identifying that sodium bicarbonate may be expected to induce more GI disturbance. This recent study implemented a 300 mg.kg-1 BM sodium citrate dose, which is smaller than is typically implemented in sodium citrate research (500 mg.kg-1), so some further investigation is still needed to compare these two dietary supplements. As astutely noted in the Peacock et al. paper, the existing evidence on GI symptoms within the sodium bicarbonate and sodium citrate literature may be confounded by the use of varying measurement approaches (questionnaires, scales etc.) across prior investigations. As such, the authors have added a comment to lines 43-49 on page 3 of the revised manuscript, but have not discussed this in substantial detail, due to the limited existing evidence.

“These dietary supplements have been reported to induce some gastrointestinal (GI) disturbances [4-6], but it has been proposed that sodium citrate may induce fewer GI symptoms than sodium bicarbonate [7, 8]. A recent investigation identified that sodium citrate was indeed associated with reduced GI disturbances compared to sodium bicarbonate when the supplements were ingested at the same dose (300 mg.kg-1 BM) [8]. These findings provide preliminary evidence that sodium citrate may be a preferred alkalising agent from a GI disturbance perspective.”

Line 46: There is a very abrupt start to this paragraph and it would benefit from either a more introductory statement, or a better link from the previous paragraph. 

Thank you for providing this suggestion. The authors have added an introductory segment to this sentence on lines 50-52 on page 3 of the revised manuscript.

“While buffering agent ingestion is typically undertaken with the intent of improving subsequent exercise performance, equivocal effects on exercise performance have been reported after sodium citrate supplementation [9-11].”

Line 47: “identified that ergogenic benefit…” should have an “an” between that and ergogenic. Line 49: move > 100% … to after “exercise”. 

Thank you for highlighting these issues. The authors have now made the recommended changes to the Introduction on lines 52-54 of page 3 of the revised manuscript.

“A recent review identified that an ergogenic benefit was more frequently reported when completing short duration (> 60 s and < 420 s) and very high-intensity exercise (> 100% VO2max) compared to longer-duration exercise of any intensity [12].”

Lines 54-55: state why this ingestion time has previously been suggested. 

The authors have addressed this question under the assumption that this comment refers to the ‘dose’ rather than ‘ingestion time’, given the line indicated by the reviewer. As such, the authors have added some detail to lines 61-63 on pages 3 and 4 of the revised manuscript.

“Based on the induced blood alkalosis and gastrointestinal symptoms reported in prior dose-response investigations, sodium citrate is recommended to be ingested at a dose of 500 mg.kg-1 body mass (BM) [4, 15].”

Line 58: The clarity of your terminology regarding the “duration of ingestion” is really confusing. The reader can easily confuse this with the pre-exercise ingestion period, so it is important for you to define exactly what you mean here. It took me a few attempts to read this section and fully understand it (this may well be a reflection on me of course, but it’s likely to also be the case for others). 

Thank you for providing this comment. The authors have accepted this suggestion, and have adjusted the terminology throughout the manuscript. The previously used “ingestion duration” is now referred to as the “ingestion period” throughout the manuscript, and a definition of this terminology has been added to lines 64-67 of the Introduction of the revised manuscript. Please also note that this change in terminology has been implemented in the manuscript title, abstract, and all figures and tables. 

“While sodium citrate dose and ingestion mode are somewhat established, no prior investigation has assessed the effect of the ingestion period (i.e. the time taken to complete ingestion of the entire dose of the supplement) on subsequent blood alkalosis, GI symptoms or palatability.”

Line 62: This feels a bit of a dead end. Where does it lead, where’s the “so what?” 

Thank you for identifying this issue. The authors have adjusted the terminology and added detail to this section, on lines 67-73 on page 4 of the revised manuscript.

“The combined effect of a specific dose, mode and period of sodium citrate supplementation may also contribute to the suggested timing of ingestion (relative to the onset of exercise) required for performance benefit. Currently, sodium citrate supplementation is recommended to take place at least 180 min before the onset of exercise [4, 16], but changed physiological responses, palatability or GI symptoms according to the duration of the ingestion period have yet to be established.”

Lines 63-67: See previous comment about clarity of this ingestion period information. 

As noted in response to the above comment, the authors have adjusted this terminology throughout the manuscript.

Line 78: The Heibel et al and Jones et al., papers might be better here. 

Thank you for this suggestions. The authors agree that the Jones et al. (2016) and Heibel et al. (2018) papers are appropriate to be cited here, and these citations are now included on Line 89 on page 5 of the revised manuscript. 

Line 81: “persistence” is an interesting choice of word, but it probably needs a bit of clarification. 

Thank you for making this suggestion. The authors have adjusted the phrasing here to be more consistent with that used in the rest of the manuscript. The change is reflected in the Introduction on lines 92-94 on page 5 of the revised manuscript.

“Extending this post-ingestion observation period to 480 min would double the time explored by prior investigations, and may be sufficient to observe the full time interval where blood alkalosis remains elevated following sodium citrate supplementation.”

Line 83: “changed” might be better as “alterations to” or something similar. 

Thank you for this suggestion. The authors have accepted this comment, which is now reflected in the Introduction on line 95 on page 5 of the revised manuscript.

“Sodium citrate supplementation likely induces blood alkalosis via alterations to strong ion difference (SID) [30, 31]…”

Line 91: Check the grammar here. 

The authors have checked the grammar as requested, and have added a comma between the last two listed variables on line 104 on page 5 of the revised manuscript. 

“Secondary aims were to establish the effect of sodium citrate ingestion period on blood [Na+], blood [Cl-], plasma [citrate], GI symptoms, and palatability.”

Methods

Line 95: Why 16? 

Thank you for this query. For the current investigation, it was not possible to perform an accurate sample size estimate, due to an absence of comparable data in the prior sodium citrate literature, however we did use all available information to determine an appropriate sample size. Relevant data for a sample size calculation would have required a prior investigation to compare sodium citrate (or bicarbonate) ingestion durations over an eight hour period with no exercise performance test, given that the current investigation was assessing resting pre-exercise responses to supplementation only. The final study sample size of 16 participants in a cross-over design where four conditions were tested exceeds that of the vast majority of prior sodium citrate investigations where meaningful results have been reported for both resting blood alkalosis and also for exercise performance. The authors have acknowledged this as a limitation on lines 467-471 on page 26 of the revised manuscript.

“Finally, the sample size used in this study was based on pragmatic considerations, due to a lack of available data from comparable investigations assessing the effect of varying sodium citrate ingestion periods. However, the final sample size of 16 participants in a cross-over design where four conditions were tested exceeds the sample sizes of the majority of prior sodium citrate investigations.”

Line 96: height should always be described in m not cm. Lines 104-105: you don’t need to state the units for height and weight, it’s obvious, and you’ve already done it. 

Thank you for highlighting this oversight. The authors have now adjusted the unit for height to metres rather than centimetres, as is reflect on line 108 on page 5 of the revised manuscript. Additionally, the units for height and body mass are no longer stated in the subsequent section (Study Design Overview), as per your suggestion. 

Lines 119-120: “10:00 pm the night prior” is a little awkwardly written. Just state the washout period between trials was… you don’t need to clarify this with two sets of terms. 

The authors have clarified the phrasing regarding the commencement of the fasting period on lines 132-133 on page 6 of the revised manuscript. This, however, does not refer to the washout period. As such, the authors have maintained the final sentence of the Study Design Overview session in its original form, where the washout between sessions is stated. 

“Participants arrived at the laboratory following an overnight fast that commenced at 10:00 pm the night before each session.”

Line123: Please state the size of the capsules used to allow for possible replication. 

Thank you for identifying this missing detail. The authors have now included the size of the capsules in the Methods section, on line 137 on page 7 of the revised manuscript.

“Participants ingested 500 mg.kg-1 BM sodium citrate (34.2 ± 6.3 capsules per participant) in size 0 gelatine capsules…”

Line 127: Why was such a small meal chosen. How does this relate to likely practice in feeding prior to exercise? 

Thank you for posing this important question. The authors propose that the meal included in the current investigation (1.75 g/kg/bm CHO), while not excessive, is also not particularly small. This amount of carbohydrate is equal to or greater than that included in prior similar investigations of sodium citrate supplementation protocols [1, 2]. Further, this amount of carbohydrate falls within the range highlighted for pre-event fuelling as recommended to athletes by the ACSM [3]. 

As an example, a participant with a body mass equal to the mean of our entire participant group (68.3 kg) was provided with a meal comprising 119.5 g of CHO. This meal consisted of 750 mL of Powerade, 1 medium sized banana (approx. 120 g), 2 slices of multi-grain bread (approx. 20 g per slice), 1 serving of strawberry conserve/jam (13.6 g), and 1 Carman’s super berry muesli bar (approx. 45 g). 

Line 135: it is unclear if this was exactly the same meal prescribed again. Please clarify. 

The authors have added this detail to lines 150-152 on page 7 of the revised manuscript.

“Participants remained seated for the next 480 min, and a second meal was provided 240 min after completion of sodium citrate ingestion, comprising the same foods, quantities and carbohydrate content as the initial co-ingested meal.”

Lines 147-148: Whilst the radiometer’s validity and reliability are well established, can you provide some estimate or support for this assay’s TEM/CV or validity and reliability. 

Thank you for providing this comment. The authors have sought to address this in the manuscript and in this response. The validity of this assay is presented in a study conducted by Moellering and Gruber (1966), the citation to which is now also included on line 164 on page 8 of the revised manuscript. 

The authors have also conducted CV and TEM analyses on our own assay data to represent the reliability of said assay. The CV values have been added to lines 164-165 on page 8 of the revised manuscript.

“The intra-assay percentage coefficient of variation (%CV) for a mid-range citrate standard solution (i.e. 250µM) was 3.4%, and the inter-assay %CV was 10.3%.”

Further, the typical error of measurement (TEM) for this standard, with a mean delta fluorescence equal to 146.5, was 3.6 for the intra-assay and 10.7 for the inter-assay variation. The authors opted not to include the TEM values in the revised manuscript to prioritise clarity for the reader. 

Line 151: add “the” between “and” and “volume”. 

Thank you for providing this suggestion. The authors have accepted this comment, and made the change which is now reflected in the revised manuscript.

Lines 164-165: This is a statement, not a paragraph. It should also be placed last in this section, as it’s not an appropriate starting statement for a statistical analysis section. 

Thank you for providing this suggestion. The authors have accepted this suggestion, and moved the statement to lines 210-211 on page 10 of the revised manuscript.

Lines 166-170: how did you determine the most appropriate model to use? What were the criteria? It would also be best practice to provide an estimate of effect size for these analyses. For the LMM you could consider Cohen’s F squared. 

Thank you for this query. The models proposed in this study were directly defined by the factors involved in the experimental design (i.e. time, treatment, interaction time and treatment, order). 

Regarding effect size estimates, we have now provided estimated differences and 95% confidence intervals for pairwise comparisons between treatments for all variables (excluding GI symptoms due to skewed data distributions) in the supplementary tables (S1, S2, S3, S4) that correspond to the tables within the revised manuscript (Table 1, Table 2, Table 3, Table 4). Pairwise comparisons have been provided for all outcomes independently of main treatment effect statistical significant to provide the reader with additional information on the effect sizes. We have also included, in the description of the results, the estimated differences instead of only p-values, so that readers can assess the magnitude of the differences for significant results. We used this approach rather than the standardised effect size measure because all of our outcomes are physiological or time (min) measures and therefore the absolute differences are clinically relevant and allow simpler interpretation. 

Line 178: Don’t use personal pronouns in scientific writing. I know it’s become fashionable of late to do this, but it’s not good practice. Please do this for the other instances of this in the manuscript (we and our) are used on multiple occasions. 

Thank you for providing this suggestion. The authors have accepted this comment and adjusted this throughout the revised manuscript. 

Line 190; Don’t start sentences with abbreviations. Check this throughout the manuscript. 

Thank you for providing this suggestion. The authors have accepted this comment and adjusted this throughout the revised manuscript. 

Lines 194-195: “variables a and b..” this needs more clarity. 

Thank you for providing this suggestion. The authors have now modified this sentence accordingly, and the changes appear on lines 209-210 on page 10 of the revised manuscript.

“The sum of all symptoms (a) and the total session rating (b) were reported as mean and range.”

Results

This is the section that I had some serious trouble with. I think it needs a complete re-think, because I got completely lost whilst reading it, and given that I have good knowledge and experience of this type of study, that means that most other will also likely struggle to access the information. The main issue is that there is just so much data and making comparison between the trial conditions is really difficult to follow. This isn’t helped by the clarity of some of the figures, which need solid lines and markers, and they need some of the error bars removing so that the actual data is visible, because it just look far too untidy at present. The key focus must surely be the comparison in the responses to the ingestion time periods, so I think you need to display all of the variable with that in mind, to allow the reader to make this visual comparisons for themselves. 

The authors thank the reviewer for this thoughtful and thorough comment. The authors have considered this comment in depth, and have made substantial changes that are described below. 

The authors acknowledge the limitations of the figure presentation as a part of the original submission and have made adjustments to all figures, which are attached to this resubmission. The major changes include combining several figures from the original submission, to reduce the total number of figures from 7 down to 3. Further, as suggested, the authors have now used solid lines rather than dashed lines on each chart, removed negative (below mean) confidence intervals (except for in Figure 1E due to the use of negative confidence interval in determining a window of time where blood [HCO3-] increase above baseline exceeded 6 mmol.L-1) to reduce clutter on each figure. 

We have also reorganised the results section as suggested, and have indicated with italicised headings the results being described in each paragraph. We have included the estimated differences between pairs of treatments along with 95% CIs for all significant comparisons so that the readers can assess the magnitude of the effect associated with each comparison. We have added supplementary tables (Tables S1, S2, S3, and S4) where we report model-based pairwise comparisons between the four ingestion periods.

I was a little unsure if the figure titles were titles, or sub-sectioned paragraphs as they seemed to be presented in the middle of the results section, which was a little confusing. 

Thank you for providing this comment. The authors would like to respectfully point out that the figure titles were formatted according to the requirements of PLOS ONE. As such, this formatting and placement has been retained from the original manuscript. With the reduction in the number of figures, and therefore figures titles, from 7 down to 3 (outlined in the previous response), there may be less confusing presented by the figure titles.

One strategy for making the description of the results easier to read, it to avoid using the term “significant” as much as possible. That forces you to write about data, rather than the stats outcomes. Remember that the stats outcomes, are there as supporting evidence, they should not be the sole focus. Try to describe the response of the variables and that show improve this section. 

Thank you for providing this suggestion. The authors have removed or changed the use of the word ‘significant’ throughout the revised manuscript, particularly in the Discussion section where the key findings are elaborated on. Further to this, we have added estimated differences alongside or instead of p-values in the Results section of the revised manuscript. 

Discussion

Generally, this was a good section with some nice flow and useful information and interpretation. 

Line 329: given the size of some of the error bars, certainly initially, I’m not convinced that all of the participants would calls their GIS as “minor”. So can you re-phrase this to reflect that. 

To ensure that the data is presented in a manner that better represents the GI symptoms, the y-axes in Figure 3A and 3B have been adjusted within the revised manuscript, so that the axis break occurs at a rating of 24 rather than 12. Further, the authors have changed the terminology on line 323-325 of page 20 of the revised manuscript, in order to more accurately report that the symptoms were overwhelmingly minor in the context of the maximum number/severity of symptoms that could have been reported by our participants.

“The sodium citrate ingestion period did not impact palatability, and the mean reported GI symptoms were minor for all ingestion protocols at all times.”

Line 336: replace “levels” with “concentration” throughout the manuscript when describing a blood or urine parameter. 

Thank you for providing this suggestion, the authors have accepted and made this change throughout the revised manuscript.

Line 338: we can assume this is significant, so you don’t need the p-value. 

Thank you for providing this suggestion, the authors have accepted and made this change to the revised manuscript.

Line 346: this is awkwardly phrased. 

Thank you for providing this comment, the authors have adjusted the phrasing on lines 345-348 on page 21 of the revised manuscript to clarify this point.

“Prior investigations had undertaken post-ingestion observation periods with a duration of up to 240 min [4, 16], therefore the complete time interval where induced blood alkalosis was significantly elevated (beyond 240 min post-ingestion) was unknown.”

Line 350: “dosage” is a poor term. Dose is better. It would be worth clarifying this type of effect, to that seen with delayed-release capsule ingestion as the response of more gradual absorption is similar. 

Thank you for providing this suggestion, “dosage” has now been changed to “dose” in the revised manuscript.

The authors have also added a section to lines 352-357 on page 21 of the revised manuscript in reference to the second portion of this comment.

“This effect may be considered similar to that previously reported within the sodium bicarbonate literature, when comparing gelatine capsules to delayed-release capsules. Delayed-release or enteric-coated capsules have been reported to delay the occurrence of peak blood alkalosis when compared to gelatine capsules (by approx. 20-24 min [43, 44]) in a similar fashion to that seen when comparing the 15 and 60 min ingestion periods in the current investigation (by approx. 20 min).”

Line 354: “our laboratory” is a bit pretentious. Is this really needed? Just reference the paper. 

Thank you for providing this suggestion. The authors have accepted this comment and adjusted this throughout the revised manuscript. 

Lines 362-368: Nice section. 

Double check you use of square parentheses throughout the manuscript, especially in sentences where you have them next to “blood”. 

The authors have proofed/edited the manuscript, and can confirm that the use of square brackets currently appears to be correct, in that square brackets have been used only to represent a concentration (e.g. blood [HCO3-] represents a concentration of bicarbonate ions within the blood).

Lines 394-396: This may have been due to capsule size differences. 

Thank you for highlighting this potential explanation. The authors have adjusted the discussion of these findings on lines 405-410 on pages 23-24 of the revised manuscript.

“This occurred one to two hours later than that seen in prior investigations of plasma [citrate] [48, 49]. These differences are most likely due to the greater citrate load ingested in the current investigation (500 mg.kg-1 BM sodium citrate dose vs 40 mEq. potassium citrate (approximately 185 mg.kg-1 BM dose)), and/or possibly due to the differing ingestion modes implemented across studies (gelatine capsules in the current investigation, tablets or solution in prior investigations [48, 49]).”

The discussion might be better with the subheading removed, and the material in each included in other paragraphs or elsewhere (just a consideration really). 

Thank you for providing this suggestion. The authors have thoroughly considered and discussed this idea, but have decided to retain the current discussion structure in terms of sub-headings and content location. The authors opine that addressing each sub-headed topic in isolation allows for deeper elaboration on each key finding of the current investigation, and the sub-headings allow for easier navigation for the reader.

Lines 431-439: This section reads like a results section, rather than a discussion. 

Thank you for this comment. The authors have adjusted the focus of this ‘Palatability’ section on lines 446-456 on pages 25 and 26 of the revised manuscript. This section now focusses more on the practicality or transferability of these findings.

 “No difference was detected when comparing the sodium citrate ingestion protocols of varying duration for palatability (p > 0.05). Palatability (across treatments) ranged from 5.3 to 5.9, which corresponds to a rating of between ‘neither like nor dislike’ and ‘like slightly’ [40]. A low palatability, especially for foods or fluids that are excessively salty, can be associated with avoidance of that particular food or fluid [53, 54]. From a practical perspective, higher palatability ratings may increase the extent to which a dietary supplementation strategy could be implemented in the context of high-intensity exercise. The broad similarity in palatability across treatments was to be expected, given the standardisation of the sodium citrate dose, ingestion mode, and co-ingested food and fluid. The moderate palatability ratings reported by participants of this investigation indicate that sodium citrate may be feasible for inclusion in training or competition routines.”

Lines 441: can you really conclude that athletes can use this strategy, given that you have recruited active participants? 

Thank you for identifying this. This point has now been addressed in the new Limitations section in the revised manuscript, which was recommended by reviewer 1. The authors have also adjusted the phrasing on lines 472-477 on page 26 of the revised manuscript.

“Based on the findings of the current investigation, it is recommended that sodium citrate be ingested across a 15-45 min period, with ingestion completed 150-270 min before the commencement of exercise, particularly for individuals competing in events where there may be performance benefit after induced blood alkalosis. Adherence to the recommended dose of 500 mg.kg-1 BM, ingested in gelatine capsules alongside a small carbohydrate-rich meal is recommended.”

Figures and Tables 

Tables: these are generally clear, but they would benefit from the removal of as many borders and lines as possible. 

Thank you for providing this suggestion, the authors have accepted and actioned this in the revised manuscript.

Figures: add line markers and remove some of the error bars for clarity. Don’t use letters to highlight significance as the figures with multiple graphs on are also labelled with letters. Make it easier for the reader to make comparisons between trials.

Thank you for providing these suggestions. The authors have added feint line markers to all figures. The authors have adjusted the indicators of significance on all parts of Figures 1 and 2, and made the related changes to the figure titles/descriptions in the revised manuscript. The updated figures are attached as a part of this re-submission. 

1. Urwin CS, Dwyer D, Carr AJ. Induced alkalosis and gastrointestinal symptoms after sodium citrate ingestion: a dose-response investigation. Int J Sport Nutr Exerc Metab. 2016;26(6):542-8.

2. Urwin CS, Snow RJ, Orellana L, Condo D, Wadley GD, Carr AJ. Sodium citrate ingestion protocol impacts induced alkalosis, gastrointestinal symptoms, and palatability. Physiological Reports. 2019;7(19):14216.

3. Thomas D, Erdman K, Burke L. American College of Sports Medicine joint position statement on nutrition and athletic performance. Med Sci Sports Exerc. 2016;48(3):543-68.

---

## [Decision Letter · Decision Letter 1]

4 May 2021

Does varying the ingestion duration of sodium citrate influence blood alkalosis and gastrointestinal symptoms?

PONE-D-21-03700R1

Dear Dr. Urwin,

We’re pleased to inform you that your manuscript has been judged scientifically suitable for publication and will be formally accepted for publication once it meets all outstanding technical requirements.

Kind regards,

Lars McNaughton, PhD

Academic Editor

PLOS ONE

Additional Editor Comments (optional):

Reviewers' comments:

Reviewer's Responses to Questions

**Comments to the Author**

1. If the authors have adequately addressed your comments raised in a previous round of review and you feel that this manuscript is now acceptable for publication, you may indicate that here to bypass the “Comments to the Author” section, enter your conflict of interest statement in the “Confidential to Editor” section, and submit your "Accept" recommendation.

Reviewer #1: All comments have been addressed

Reviewer #2: All comments have been addressed

2. Is the manuscript technically sound, and do the data support the conclusions?

Reviewer #1: Yes

Reviewer #2: Yes

3. Has the statistical analysis been performed appropriately and rigorously? 

Reviewer #1: Yes

Reviewer #2: Yes

4. Have the authors made all data underlying the findings in their manuscript fully available?

Reviewer #1: No

Reviewer #2: Yes

5. Is the manuscript presented in an intelligible fashion and written in standard English?

Reviewer #1: Yes

Reviewer #2: Yes

6. Review Comments to the Author

Reviewer #1: I would like to thank the authors for considering my comments, which I believe they have adequately addressed.

Congratulations on the interesting work, looking forward to more!

Reviewer #2: The authors should be commended on the high volume of work, along with the quality of what they have done in revising the manuscript. It reads far better than the original submission and you have addressed all of my comments and suggestions.

7. PLOS authors have the option to publish the peer review history of their article (what does this mean?). If published, this will include your full peer review and any attached files.

Reviewer #1: **Yes: **Bryan Saunders

Reviewer #2: **Yes: **S. Andy Sparks

---

## [Editor Report · Acceptance letter]

7 May 2021

PONE-D-21-03700R1 

Does varying the ingestion period of sodium citrate influence blood alkalosis and gastrointestinal symptoms? 

Dear Dr. Urwin:

I'm pleased to inform you that your manuscript has been deemed suitable for publication in PLOS ONE. Congratulations! Your manuscript is now with our production department. 

Kind regards, 

on behalf of

Dr. Lars McNaughton 

Academic Editor

PLOS ONE